# D$_1$- and D$_2$-like receptors differentially mediate the effects of dopaminergic transmission on cost–benefit evaluation and motivation in monkeys

**Yukiko Hori**[1], **Yuji Nagai**[1], **Koki Mimura**[1], **Tetsuya Suhara**[1], **Makoto Higuchi**[1], **Sebastien Bouret**[2], **Takafumi Minamimoto**[1]*

**1** Department of Functional Brain Imaging, National Institutes for Quantum and Radiological Science and Technology, Chiba, Japan, **2** Team Motivation Brain & Behavior, Institut du Cerveau et de la Moelle épinière (ICM), Centre National de la Recherche Scientifique (CNRS), Hôpital Pitié Salpêtrière, Paris, France

* minamimoto.takafumi@qst.go.jp

**Data Availability Statement:** All data presented in this paper have been posted on the following public

## Abstract

It has been widely accepted that dopamine (DA) plays a major role in motivation, yet the specific contribution of DA signaling at D$_1$-like receptor (D$_1$R) and D$_2$-like receptor (D$_2$R) to cost–benefit trade-off remains unclear. Here, by combining pharmacological manipulation of DA receptors (DARs) and positron emission tomography (PET) imaging, we assessed the relationship between the degree of D$_1$R/D$_2$R blockade and changes in benefit- and cost-based motivation for goal-directed behavior of macaque monkeys. We found that the degree of blockade of either D$_1$R or D$_2$R was associated with a reduction of the positive impact of reward amount and increasing delay discounting. Workload discounting was selectively increased by D$_2$R antagonism. In addition, blocking both D$_1$R and D$_2$R had a synergistic effect on delay discounting but an antagonist effect on workload discounting. These results provide fundamental insight into the distinct mechanisms of DA action in the regulation of the benefit- and cost-based motivation, which have important implications for motivational alterations in both neurological and psychiatric disorders.

## Introduction

In our daily lives, we routinely determine whether to engage or disengage in an action according to its benefits and costs: The expected value of benefits (i.e., rewards) has a positive influence, while the cost necessary to earn the expected reward (e.g., delay, risk, or effort) decreases the impact of reward value [1–3]. Arguably, the dopamine (DA) system plays a central role in the motivation, which adjusts behavior as a function of expected costs and benefits. Phasic firing of midbrain DA neurons positively scales with the magnitude of future rewards and negatively scales with risk or time delay to reward [4–11]. In addition, several studies demonstrated that DA neurotransmission was causally involved in regulation of behavior based on expected costs and benefits [12–18]. In patients suffering from

repository: https://github.com/minamimoto-lab/
2021-Hori-DAR.

**Funding:** This research was supported in part by
KAKENHI Grants JP26282221, JP26120733,
JP18H04037, and JP20H05955 from Japan
Society for the Promotion of Science (JSPS)
(http://www.jsps.go.jp/english/index.html) to TM,
and by Japan Agency for Medical Research and
Development (AMED) (https://www.amed.go.jp/en/
index.html) Grant Numbers JP20dm0107094 (to
TS). The funders had no role in study design, data
collection and analysis, decision to publish, or
preparation of the manuscript.

**Competing interests:** The authors have declared
that no competing interests exist.

**Abbreviations:** 3D, three-dimensional; BIC,
Bayesian information criterion; CON, control; CU,
cost unit; $D_1R$, $D_1$-like receptor; $D_2R$, $D_2$-like
receptor; DA, dopamine; DAR, DA receptor; dlPFC,
dorsolateral prefrontal cortex; HO, high occupancy;
HSD, honestly significant difference; i.m.,
intramuscular; LMM, linear mixed model; MO,
moderate occupancy; MR, magnetic resonance;
NAcc, nucleus accumbens; PD, Parkinson disease;
PET, positron emission tomography; RT, reaction
time; vlPFC, ventrolateral prefrontal cortex; VOI,
volume of interest.

depression, schizophrenia, or Parkinson disease (PD), the alteration of DA transmission is frequently associated with various pathological impairments of motivation such as anergia, fatigue, psychomotor retardation, and apathy [14,19–21]. DA signaling is mediated at post-synaptic sites by 2 classes of DA receptors (DARs), the $D_1$-like receptor ($D_1R$) and the $D_2$-like receptor ($D_2R$), and both classes are thought to be involved in the regulation of motivation [22,23].

However, the specific mechanisms through which DA contributes to motivation based on cost–benefit trade-off remain unclear. For example, in tasks where animals must exert a higher force to obtain a bigger reward, blockade of either $D_1R$ or $D_2R$ shifts preferences toward less efforts, thus less rewards, suggesting a role of DA in effort [24–29]. On the other hand, since DA activity shows little sensitivity to information about effort when it is decoupled from reward, it has been proposed that DA is strongly involved in adjusting motivation based on expected benefits (reward availability) rather than on expected energetic costs (effort) [9,30,31]. Note that this apparent controversy might be related to the difficulty of interpreting results from experiments where the nature of costs and benefits was not clearly identified and isolated [11].

To understand the role of DA in motivation, it is critical to identify not only the pattern of DA activity and release across costs and benefits, but also the action of DA on DARs [17]. However, the relative implication of distinct receptor subtypes in specific aspects of the cost–benefit trade-off in motivation also remains under debate. For example, systemic administration of $D_1R$ or $D_2R$ antagonist was shown to increase preference for small immediate rewards over larger, delayed rewards [25,32–34]. Some of these studies, however, have also shown that blockade of $D_1R$ [34] or $D_2R$ [33] has no effect on delay cost. These and other previous behavioral pharmacology studies have compared the effect of DAR blockade according to the antagonist dose–response relationship for each DAR subtype. However, since different antagonists display distinct pharmacological properties (e.g., target affinity, brain permeability, and biostability), it is difficult to accurately predict the effects on their target receptors in vivo. Therefore, to describe the role of DARs in motivational processes beyond a simple dose–response relationship, it seems essential to measure receptor occupancy after antagonist administration. Indeed, positron emission tomography (PET) studies of patients have shown that in vivo $D_2R$ occupancy is a reliable predictor of clinical and side effects of antipsychotic drugs [35,36]. Similarly, receptor occupancy has been measured in rats and monkeys, as well as the relationship with the behavioral effects following $D_2R$ antagonists [37–39].

In the present study, we aimed to quantify and directly compare the roles of DA signaling via $D_1R$ and $D_2R$ in motivation based on the costs and benefits in macaque monkeys. For this purpose, we manipulated DA transmission by systemic injections of specific antagonists for $D_1R$ and $D_2R$ and assessed the degree of DAR occupancy using in vivo PET imaging with selective radioligands. The effects of this quantitatively controlled DAR blockade on benefit- and cost-based motivation were evaluated in 2 sets of behavioral experiments. First, we quantified the effects of DAR blockade on the incentive impact of reward prediction, namely the relationship between predicted reward amount and the motivation of a goal-directed task. Second, to assess the effect of DAR blockade on 2 types of costs, workload and delay, we used a similar behavioral task for a fixed amount of reward, but either cost was implemented, allowing us to estimate the negative impacts of cost as steepness of reward discounting (i.e., workload and delay discounting). Based on our data, $D_1R$ and $D_2R$ have similar roles in incentive impact of reward prediction and delay discounting, whereas workload discounting is exclusively related to $D_2R$ manipulation.

## Results

### PET measurement of $D_1R/D_2R$ occupancy following systemic antagonist administration

To establish appropriate antagonist doses and experimental timing, we measured the degree of receptor blockade (i.e., receptor occupancy) following systemic administration of DAR antagonists. We performed PET imaging with selective radioligands for $D_1R$ ([$^{11}$C]SCH23390) and $D_2R$ ([$^{11}$C]raclopride) in a total of 4 monkeys (3 for each) under awake condition for both baseline (without drug administration) and following antagonist administration. We quantified specific radioligand binding using a simplified reference tissue model with the cerebellum as reference region.

For $D_1R$ measurement, high radiotracer binding was seen in the striatum at baseline condition (Fig 1A, baseline). PET scans were obtained after pretreatment with non-radiolabeled SCH23390 for $D_1R$ antagonist at different doses (10, 30, 50, and 100 μg/kg), demonstrating that specific tracer binding was diminished in a dose-dependent manner (Fig 1A). We performed a volume of interest (VOI)-based analysis quantifying the reduction of specific bindings from baseline, which was homogenous across several brain regions within a blocking condition (S1 Fig). We defined receptor occupancy as the degree of reduction of specific binding using the values from striatal VOI, since they appeared to be the most reliable (see Materials and methods) [40]. In 3 monkeys, we measured the relationship between $D_1R$ occupancy and the dose of SCH23390, which was approximated by a Hill function (Fig 1C and Eq 4). We

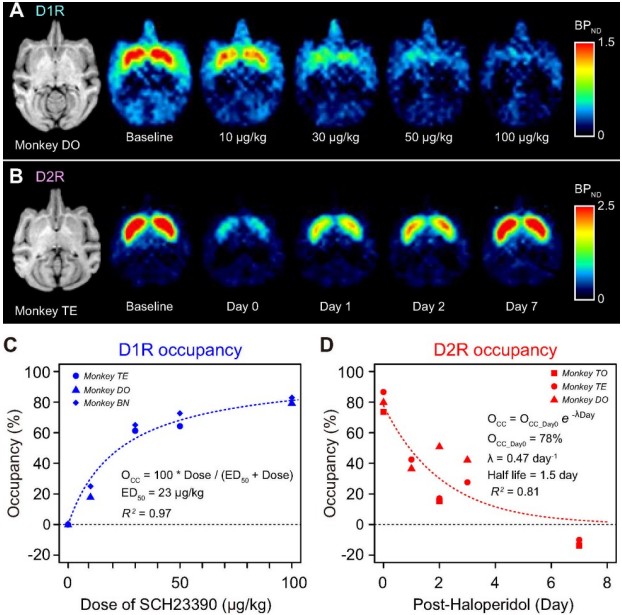

**Fig 1. $D_1R$ and $D_2R$ occupancy measured by PET. (A)** Representative horizontal MR (left) and parametric PET images showing specific binding ($BP_{ND}$) of [$^{11}$C]SCH23390 at baseline and following drug treatment with SCH23390 (10, 30, 50, or 100 μg/kg, i.m.). **(B)** Representative horizontal MR (left) and parametric PET images showing specific binding ($BP_{ND}$) of [$^{11}$C]raclopride at baseline and on 0 to 7 days after injection with haloperidol (10 μg/kg, i.m.). Color scale indicates $BP_{ND}$ (regional binding potential relative to non-displaceable radioligand). **(C)** Occupancy of $D_1R$ measured at striatal ROI is plotted against the dose of SCH23390. Three of 4 doses were examined in each monkey. **(D)** Occupancy of $D_2R$ measured at striatal ROI is plotted against the day after haloperidol injection. Dotted curves in C and D are the best fit of Eqs 4 and 5, respectively. The data underlying this figure can be found on the following public repository: https://github.com/minamimoto-lab/2021-Hori-DAR. $D_1R$, $D_1$-like receptor; $D_2R$, $D_2$-like receptor; i.m., intramuscular; MR, magnetic resonance; PET, positron emission tomography; ROI, region of interest.

found that treatment with SCH23390 at doses of 100 and 30 μg/kg corresponded to 81% and 57% of $D_1R$ occupancy, respectively.

Haloperidol was used for $D_2R$ antagonism. Unlike SCH23390, which was rapidly washed from the brain within a few hours, a single dose of haloperidol treatment was expected to show persistent $D_2R$ occupancy for the following several days as described in humans and mice [41,42], providing the opportunity to test different occupancy conditions. The baseline [$^{11}$C]raclopride PET image showed the highest radiotracer binding in the striatum (Fig 1B, baseline). As expected, striatal binding was diminished not only just after pretreatment with haloperidol (10 μg/kg, intramuscular [i.m.]), but also on post-haloperidol day 2 (Fig 1B, day 2). Binding had returned to the baseline level by day 7 (Fig 1B, day 7). We measured $D_2R$ occupancy on days 0, 1, 2, 3, and 7 after a single haloperidol injection in 3 monkeys. An exponential decay function approximated the relationship between $D_2R$ occupancy and post-haloperidol days (Eq 5); a single injection of haloperidol yielded 78% and 48% of $D_2R$ occupancy on days 0 and 1, respectively (Fig 1D).

## Effects of $D_1R$ and $D_2R$ blockade on behavior

We next quantified the effects of DAR blockade on behavior using a total of 3 monkeys not used in the PET occupancy study (monkeys KT, ST, and MP; the first 2 for incentive and all 3 for cost-based motivation, respectively). Our goal here was to study the influence of $D_1R$ and $D_2R$ manipulation on how monkeys adjusted their behavior based on expected benefits (reward size) or expected costs (delay or workload). We use tasks where reward could be obtained by performing a simple action (releasing a bar). In each version of the task, we manipulated costs (delay or workload) or benefits (reward size), such that distinct trials corresponded to different levels of cost or benefits. At the beginning of each trial, a visual cue provided information about the current cost and benefit, so that monkeys could adjust their behavior accordingly. We evaluated motivational processes by using computational modeling to measure the impact of incentive or costs on 2 behavioral measures: refusal rate (whether monkeys accepted or refused to perform the offered option; see below) and reaction time (RT; how quickly they respond).

## Effects of $D_1R$ and $D_2R$ blockade on benefit-based motivation

To assess the effect of blockade of $D_1R$ and $D_2R$ on benefit-based motivation, we tested 2 monkeys with a reward-size task (Fig 2A). In every trial of this task, the monkeys were required to release a bar when a visual target changed from red to green to get a liquid reward. A visual cue indicated the amount of reward (1, 2, 4, or 8 drops) at the beginning of each trial (Fig 2A). All monkeys had been trained to perform basic color discrimination trials in the cued multi-trial reward schedule task [43] for more than 3 months. As in previous experiments using a single option presentation, the action was very easy, and monkeys could not fail if they actually tried to release the bar on time (the error rate is indeed much lower in the absence of information about costs and benefits) [2,44]. As in those previous experiments manipulating information regarding costs and benefits, failures (either releasing the bar too early or too late) were usually observed in small reward trials and/or close to the end of daily sessions. Therefore, they were regarded as trials in which the monkeys refused to release the bar, presumably because they were not sufficiently motivated to correctly release the bar (i.e., refusal) [2]. Hence, the refusal rate provided a reliable measure of the influence of motivation on behavior [9,45–48]. We had previously shown that the refusal rate ($E$) was inversely related to reward size ($R$), which had been formulated with a single free parameter $a$ [2] (Fig 2B),

$$E = 1/aR \tag{1}$$

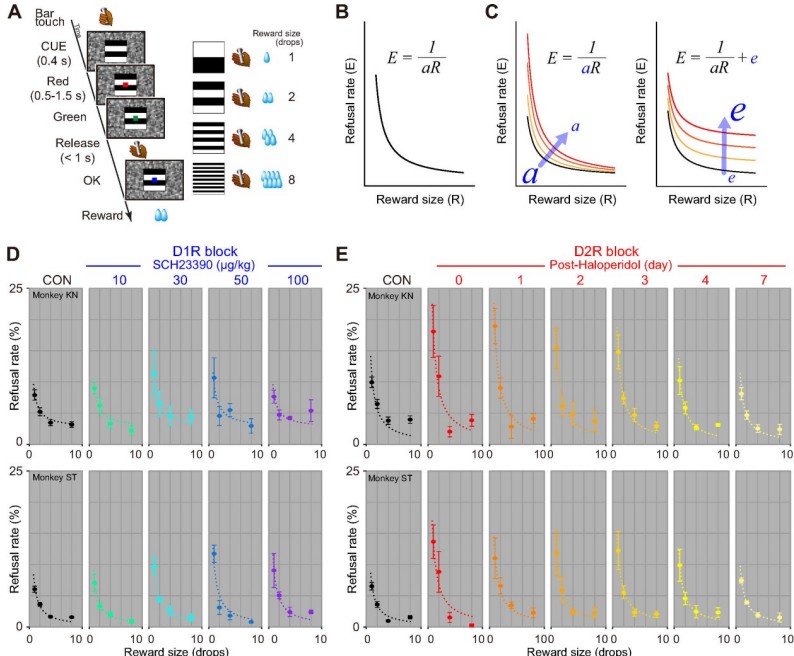

**Fig 2. $D_1R/D_2R$ blockade increased refusal rates in reward-size task. (A)** Reward-size task. Left: sequence of events during a trial. Right: association between visual cues and reward size. **(B)** Schematic illustration of inverse function between refusal rate and reward size. **(C)** Schematic illustration of 2 explanatory models of decrease in motivation. Left: increase in refusal rate (i.e., decrease in motivation) in relation to reward size caused by decrease in incentive impact ($a$). Right: an alternative model explaining increase in refusal rate irrespective of reward size. **(D, E)** Behavioral data under $D_1R$ and $D_2R$ blockade, respectively. Refusal rates (mean ± SEM) as a function of reward size for monkeys KN (top) and ST (bottom). Dotted curves are the best-fit inverse function (S1 Table). The data underlying this figure can be found on the following public repository: https://github.com/minamimoto-lab/2021-Hori-DAR. CON, control; $D_1R$, $D_1$-like receptor; $D_2R$, $D_2$-like receptor.

In agreement with these previous studies, both monkeys exhibited the inverse relationship in nontreatment condition (Fig 2D and 2E, control).

For $D_1R$ blockade, the monkeys were tested with the task 15 minutes after a systemic injection of SCH23390 (10, 30, 50, and 100 μg/kg) or vehicle as control. $D_1R$ blockade increased the refusal rates particularly in smaller reward-size trials (Fig 2D). We considered whether this increase was due to a reduction in the incentive impact of reward or a decrease in motivation irrespective of reward size. These factors can be captured by a decrease in parameter $a$ of the inverse function and implementing intercept $e$, respectively (Fig 2C). To quantify the increases in refusal rate, we compared 4 models while considering these 2 factors as random effects: model #1, random effect on $a$; model #2, random effect on $a$ with fixed $e$; model #3, fixed $a$ with random effect on $e$; model #4, random effect on both $a$ and $e$ (see S1 Table). For both monkeys, the increases in refusal rate were explained by a decrease in parameter $a$ due to the treatment, while the inverse relation with reward size was maintained (Fig 2D and S1 Table; model #2 for monkey KN and model #1 for ST). We then assessed changes in parameter $a$, which indicates the incentive impact of reward size. As shown in Fig 3A, normalized $a$ decreased as the dose of SCH23390 was increased to 30 or 50 μg/kg, but then it increased at the highest dose (100 μg/kg) for monkeys KN but less clearly so for monkey ST (Fig 3A, left).

For $D_2R$ blockade, the monkeys were tested with the task 15 minutes after a single injection of haloperidol (10 μg/kg, i.m., day 0), and they were then successively tested on the following

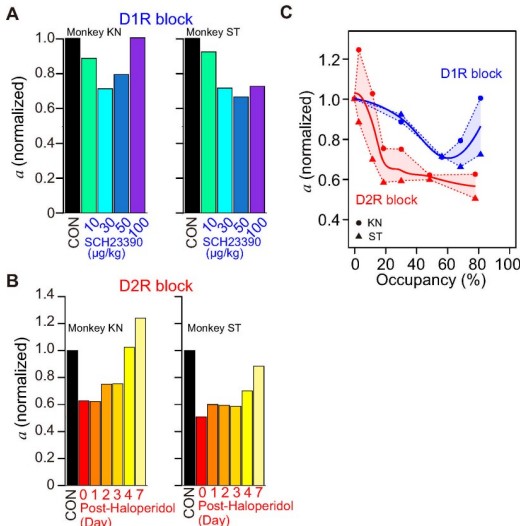

**Fig 3. Effect of $D_1R$/$D_2R$ blockade on incentive impact of reward size. (A)** Bars indicate normalized incentive impact (*a*) for each treatment condition under $D_1R$ blockade for monkeys KN and ST. The value was normalized by the value of control condition. **(B)** Same as A, but for $D_2R$ blockade. **(C)** Relationship between an incentive impact and occupancy for $D_1R$ (blue) and $D_2R$ blockade (red). Thick curves indicate LOESS of individual data (filled circles and triangles for monkeys KN and ST, respectively). The data underlying this figure can be found on the following public repository: https://github.com/minamimoto-lab/2021-Hori-DAR. $D_1R$, $D_1$-like receptor; $D_2R$, $D_2$-like receptor; LOESS, locally weighted smoothing.

days 1, 2, 3, 4 and 7. We also found an increase in refusal rates for $D_2R$ blockade in both monkeys: The refusal rates were highest on the day of haloperidol injection, after which they decreased over days (Fig 2E). Similar to the $D_1R$ blockade, the increases in refusal rate due to $D_2R$ blockade were explained solely by a decrease of parameter *a* according to the days following the treatment for both monkeys (Fig 2E and S1 Table; model #1 for both monkeys KN and ST). Our model-based analysis revealed that *a* decreased about 40% on the day of haloperidol injection and the following 3 days as compared to control and then recovered to almost the control level by day 7 (Fig 3B).

To compare the effects between $D_1R$ and $D_2R$ blockades directly, we plotted changes in incentive impact along with the degree of blockage that was normalized across 3 monkeys (Fig 3C). In both $D_1R$ and $D_2R$ blockade experiments, *a* declined according to the increase in occupancy; it gradually declined as $D_1R$ occupancy increased, but then increased at the highest occupancy, presenting a U-shaped tendency, whereas it steeply declined until 20% $D_2R$ occupancy and then continued to decrease slightly until 80% occupancy (Fig 3C). At 20% to 80% occupancy, the incentive impacts for $D_2R$ blockade stayed lower than those for $D_1R$, suggesting a stronger sensitivity of incentive impact to $D_2R$ blockade.

We sought to verify that the effect of $D_2R$ antagonism was not specific for haloperidol and also to validate the comparison between $D_1R$ and $D_2R$ in terms of receptor occupancy. We examined the behavioral effect of another $D_2R$ antagonist, raclopride, at a dose yielding about 50% receptor occupancy (10 μg/kg, i.m.; S2A Fig). Following this dose of raclopride administration in one monkey, refusal rates increased, which was explained by inverse function with $a = 5.2$ (drop$^{-1}$), a value very similar to that observed at 50% $D_2R$ occupancy with haloperidol [$a = 5.4$ (drop$^{-1}$), day 1; S2B Fig]. Thus, the reduction of incentive impact (captured by a decrease in *a* parameter) was clearly associated with the degree of $D_2$ receptor blockade regardless of the antagonist used.

## Effects of $D_1R$ and $D_2R$ blockade on response speed

To evaluate the extent to which the influence of DAR manipulation in the reward-size task could affect another behavioral measure through a single motivational process, we examined RT modulations across trials. Consistent with previous studies using systemic administration of $D_1R$ or $D_2R$ antagonists (e.g., [49]), DAR blockade prolonged RTs in a treatment-dependent manner. For $D_1R$ blockade, RTs were increased according to the antagonist dose (2-way ANOVA, main effect of treatment, $p < 1.0 \times 10^{-13}$ for both monkeys, e.g., S3A–S3C Fig, see details in legend). $D_1R$ antagonism also tended to increase the proportion of late release (2-way ANOVA, main effect of treatment, $p = 0.08$, monkey KN; $p = 0.0038$ monkey ST, e.g., S3D Fig). A simple account of these effects of $D_1R$ manipulation on RT is that the modulations in RT across conditions are caused by changes in motivation, such that the positive impact of reward on behavior affects both whether monkeys perform the action (refusal rate) as well as how quickly they will respond (RT). We reasoned that, if this were the case, then the intersession variability in RT and refusal rate should be correlated. A session-by-session analysis revealed that there was indeed a significant linear relationship between refusal rates and RTs in both monkeys, even when the treatment conditions were changed (Fig 4A and S2 Table). $D_2R$ blockade also prolonged RTs (main effect of treatment, $p < 1.0 \times 10^{-4}$, e.g., S3E–S3G Fig). $D_2R$ blockade did not change the refusal patterns (i.e., too early or late release) (2-way ANOVA, treatment, $p = 0.31$, e.g., S3H Fig). As with the case of $D_1R$, there was a linear relationship between refusal rates and RTs across $D_2R$ antagonism sessions, in which treatment had no discernible effect on the steepness of the slope (Fig 4B and S2 Table). Collectively, these

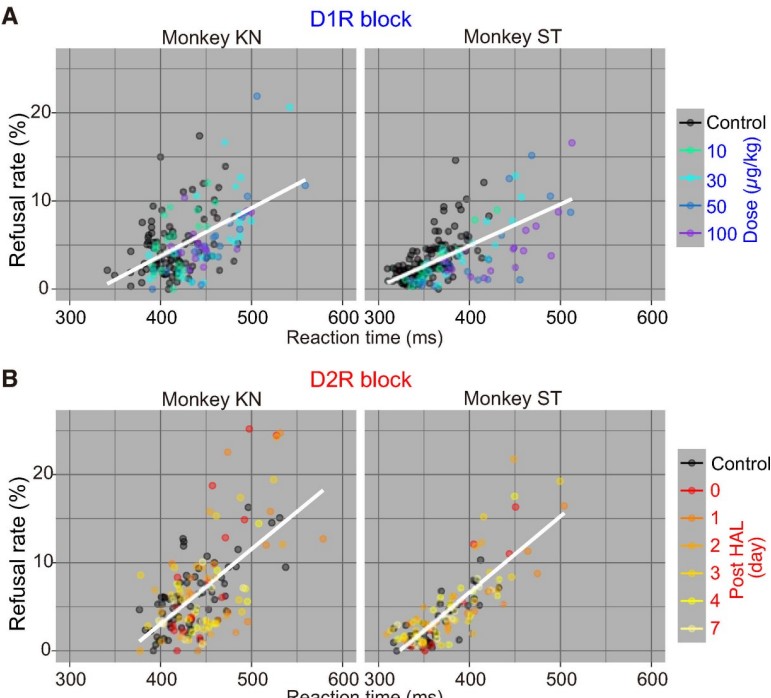

**Fig 4. Relationship between refusal rate and RT in reward-size task. (A)** Relationship between refusal rate and average RT for each reward size in session by session for $D_1R$ blocking in monkeys KN and ST. Colors indicate treatment condition. **(B)** Same as A, but for $D_2R$ blocking. Note that a simple linear regression (white line, model #1 in S2 Table) was selected as the best-fit model to explain the data. The data underlying this figure can be found on the following public repository: https://github.com/minamimoto-lab/2021-Hori-DAR. $D_1R$, $D_1$-like receptor; $D_2R$, $D_2$-like receptor; RT, reaction time.

results indicate that refusal rate and RT were similarly affected by DAR manipulation, in line with the concept that DAR affects a central process (motivation), which controls the influence of expected reward on both action selection and execution.

## Little influence of $D_1R$ or $D_2R$ blockade on hedonic impact of reward

The behavioral data shown above suggest that blockade of DAR attenuates the incentive effect of reward on behavior. To evaluate the impact of DAR blockade on other aspects of motivation and reward processing, we also examined the effect of DAR manipulation on hedonic processes, i.e., on how pleasant was the reward consumption. In line with previous experiments in rodents [15,50], we did not find any effect of treatment with $D_1R$ or $D_2R$ antagonist on overall intake or sucrose preference in either of the 2 monkeys tested (S4A Fig; see legend). We also assessed blood osmolality, a physiological index of dehydration and thirst drive [51], before and after the preference test. Again, DAR treatment had no significant influence on overall osmolality or recovery of osmolality (rehydration) (S4B Fig; see legend). These results suggest that DAR blockade has no influence on hedonic impact of reward. These results also support the notion that the increased refusal rate was not directly due to a reduction of thirst drive.

In short, these results indicate that both $D_1R$ and $D_2R$ are involved in incentive motivation, i.e., in the positive influence of the expected reward size on behavior (refusal rate and RT), but not in the hedonic impact of reward. We next examined the influence of DAR manipulation on cost processing.

## Differential effects of $D_1R$ and $D_2R$ blockades on workload and delay discounting

The trade-off between the reward and costs of obtaining the reward affects decision-making as well as motivation. Both humans and animals have the tendency to prefer immediate, smaller rewards over larger, but delayed rewards. The preference can be predicted by discounting the reward's intrinsic value by the duration of the expected delay, an effect designated as "delay discounting" [52,53]. Discounting of the reward value also occurs in proportion to the predicted effort needed to obtain the rewards, an effect called "effort discounting" [54]. Delay and effort discounting are typically measured in choice tasks, providing the relative impact of costs on reward in decision-making. Previously, we measured the discounting effect of these costs on outcome value by quantifying the relation between the amount of expected cost and the change in operant, reward-directed behavior [55].

In this study, we used the same procedure to assess the effect of selective DAR blockade on cost-based motivation. For this purpose, we used a work/delay task (Fig 5A), where the basic features were the same as those in reward-size task. There were 2 trial types. In the work trials, the monkeys had to perform 0, 1, or 2 additional instrumental actions to obtain a fixed amount of reward, and the cost (workload) scaled with the number of trials to perform. In the delay trials, after the monkeys correctly performed one instrumental trial, a reward was delivered 0 to 7 seconds later, such that the cost (delay) scaled with the time between action and reward delivery. Note that here, as in most natural conditions, greater workload is inherently associated with longer delays. Thus, in an attempt to isolate the effort component, we adjusted the delay for reward in delay trials based on the duration of corresponding workload trials: Since the timing of the trials is matched between workload and delay trials, they only differed in the number of actions and therefore in the amount of effort. At the beginning of each trial, the cost (workload or delay) was indicated by a visual cue that lasted throughout the trial. As with the reward task, we used computational modeling to quantify the influence of cost information on behavior. We have shown that the monkeys exhibited linear relationships between refusal

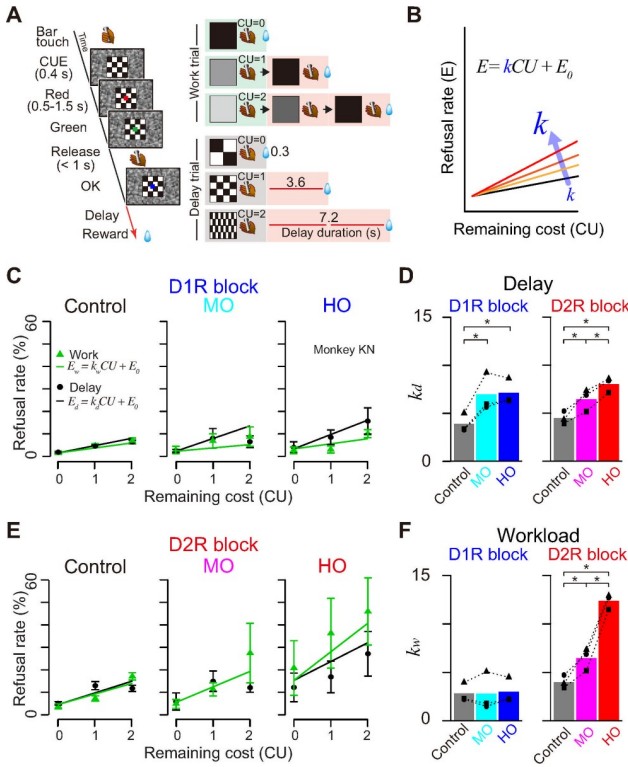

**Fig 5. Differential effects of $D_1R$ and $D_2R$ blockade on cost-based motivational valuation. (A)** The work/delay task. The sequence of events (left) and relationships between visual cues and trial schedule in the work trials (right top 3 rows) or delay duration in the delay trials (right bottom 3 rows) are shown. CU denotes the remaining (arbitrary) cost unit to get a reward, i.e., either remaining workload to perform trial(s) or remaining delay periods. **(B)** Schematic illustration of an explanatory model of increases in refusal rate by increasing cost sensitivity ($k$). **(C)** Effects of $D_1R$ blockade. Representative relationships between refusal rates (monkey KN; mean ± SEM) and remaining costs for workload (green) and delay trials (black). Saline control (Control) and moderate (30 μg/kg; MO) and high $D_1R$ occupancy treatment condition (100 μg/kg; HO) are shown. Green and black lines are the best-fit lines for work and delay trials in model #1 in S3 Table, respectively. **(D)** Effects of $D_2R$ blockade. Nontreatment control (Control), moderate (1 day after haloperidol; MO) and high $D_2$ occupancy treatment conditions (day of haloperidol; HO) are shown. Others are the same for C. **(E)** Comparison of effects between $D_1R$ and $D_2R$ blockade on delay discounting parameter ($k_d$). Bars and symbols indicate mean and individual data, respectively. **(F)** Comparison of effects between $D_1R$ and $D_2R$ blockade on workload discounting parameter ($k_w$). Asterisks represent significant difference (* $p < 0.05$, 1-way ANOVA with post hoc Tukey HSD test). The data underlying this figure can be found on the following public repository: https://github.com/minamimoto-lab/2021-Hori-DAR. $D_1R$, $D_1$-like receptor; $D_2R$, $D_2$-like receptor; HO, high occupancy; HSD, honestly significant difference; MO, moderate occupancy.

rate ($E$) and remaining costs ($CU$) for both work and delay trials, as follows:

$$E = kCU + E_0 \tag{2}$$

where $k$ is a coefficient and $E_0$ is an intercept [55] (Fig 5B). By extending the inference and formulation of reward-size task (Eq 1), this linear effect proposes that the reward value is hyperbolically discounted by cost, where the coefficient $k$ corresponds to discounting factors.

We tested 3 monkeys (monkeys KN, MP, and ST) and measured refusal rate to infer delay and workload discounting. We confirmed that refusal rates of control condition increased as the remaining cost increased (e.g., Fig 5C, control). Fig 5B illustrates our hypothesis that DAR blockade increases cost sensitivity (i.e., discounting factor, $k$), which appears as an increase in refusal rate relative to remaining cost.

                                                      

To compare the effect of $D_1R$ versus $D_2R$ antagonism on cost sensitivity at the same degree of receptor blockade, we assessed the performance of the monkeys under 2 comparable levels of DAR occupancy for $D_1R$ and $D_2R$: 50% occupancy (called "moderate occupancy" or MO) and 80% occupancy ("high occupancy" or HO). We also measured performance in absence of treatment as control. According to the occupancy study (Fig 1), MO and HO conditions corresponded to pretreatment with 30 and 100 μg/kg of SCH23390 for $D_1R$ and 1 day after and the day of haloperidol treatment for $D_2R$, respectively. Linear mixed models (LMMs) analysis verified the assumption that DAR blockade increased delay and workload discounting independently without considering the random effect of treatment condition or subject (Fig 5CD and S3 Table; see Materials and methods). We found that delay discounting was significantly increased according to the degree of DAR blockade irrespective of receptor subtype ($D_1$, $F_{(2, 4)}$ = 36.9, $p$ = 0.0026; $D_2$, $F_{(2, 4)}$ = 41.4, $p$ = 0.0021; Fig 5E). Workload discounting ($k_w$), on the other hand, was specifically increased by $D_2R$ blockade in an occupancy-dependent manner (1-way ANOVA, main effect of occupancy; $D_1$, $F_{(2, 4)}$ = 0.125, $p$ = 0.89; $D_2$, $F_{(2, 4)}$ = 243.2, $p$ = $6.6 \times 10^{-5}$; Fig 5F).

In line with what we found in the reward-size task, $D_1R$ blockade did not have any significant effect on the linear relation between refusal rate and RT in either trial type (Fig 6A). Thus, the influence of $D_1R$ manipulation on behavior could readily be accounted for by a single variable, which affects both RT and refusal rate. By contrast, $D_2R$ blockade produced an occupancy-dependent increase in the steepness of the linear relation between RT and refusal rate in workload trials, but not in delay trials (Fig 6B). Thus, $D_2R$ manipulation had a distinct influence on RT and refusal rate in workload trials, suggesting that it was acting on behavior through a distinct motivational process such as overcoming effort costs (see Discussion).

## Joint influences of $D_1R$ and $D_2R$ blockades on motivation

Considering the direct and indirect striatal output pathways where neurons exclusively express $D_1R$ and $D_2R$, respectively, and the potential functional opposition between these pathways [56], we examined the effect of joint blockade of $D_1R$ and $D_2R$. To facilitate the comparison of the influence of 2 receptors, we examined the behavioral effects of both $D_1R$ and $D_2R$ blockades at the same occupancy level. After treatment with both SCH23390 (100 μg/kg) and haloperidol (10 μg/kg), seemingly achieving approximately 80% of occupancy for both subtypes (Fig 1C and 1D), all monkeys virtually stopped performing the task: They only performed 1% to 13% of the trials compared to control conditions. When we treated the monkeys with SCH23390 (30 μg/kg) on the day following that of haloperidol injection (i.e., both $D_1R$ and $D_2R$ assumed to be occupied at approximately 50%), the monkeys had higher refusal rates in delay trials than control (Fig 7A, $D_1R$+$D_2R$ block), such that discounting factor ($k_d$) became significantly higher than in control conditions ($p < 0.05$, Tukey honestly significant difference [HSD] test; Fig 7B, delay). By contrast, this simultaneous $D_1R$ and $D_2R$ blockade appeared to attenuate the effect of $D_2R$ antagonism on workload: The refusal rates in work trials were not as high as in $D_2R$ blockade alone (Fig 7A), and the difference in workload discounting factor ($k_w$) between treated and control or baseline conditions disappeared ($p > 0.05$; Fig 7B, workload). A similar tendency of counterbalancing influences of $D_1R$ and $D_2R$ blockade was also seen in the motivation for minimum cost trials ($E_0$) (Fig 7B). These results suggest that blocking both receptor subtypes tends to induce a synergistic effect on delay discounting, while their effects on workload discounting cancel each other out.

Finally, since workload trials revealed a potential specific action of $D_2R$, with a dissociation between refusal rate and RT effects, we examined the joint influence of $D_1R$ and $D_2R$ manipulations on the relation between these 2 behavioral measures. As shown in S5B Fig, when $D_1R$

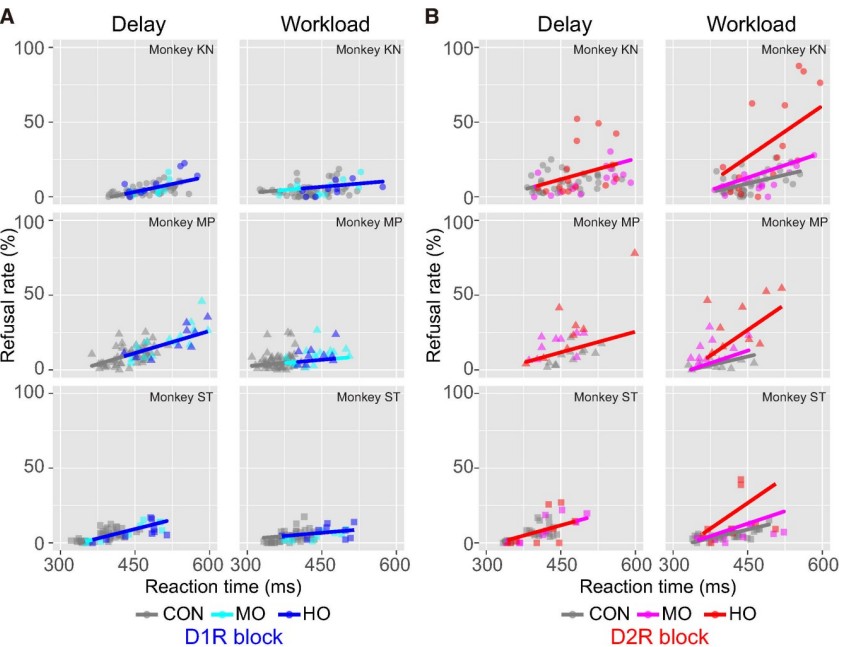

**Fig 6. Relationship between refusal rate and RT in work/delay task. (A)** Relationship between refusal rate and average RT for each remaining cost in session by session for $D_1R$ blocking in delay and workload trials. Data are plotted individually for monkeys KN, MP, and ST, in order from top to bottom. Colors indicate treatment condition. Thick lines indicate linear regression lines. **(B)** Same as A, but for $D_2R$ blocking. Note that for the data in workload trials under $D_2R$ treatment, a linear model with random effect of condition (model #4 in S4 Table) was chosen as the best model to explain the data, whereas for the other data, a simple linear regression model (model #1, without any random effect or model #2 with random effect of subject) was selected. The data underlying this figure can be found on the following public repository: https://github.com/minamimoto-lab/2021-Hori-DAR. CON, control; $D_1R$, $D_1$-like receptor; $D_2R$, $D_2$-like receptor; HO, high occupancy; MO, moderate occupancy; RT, reaction time.

and $D_2R$ were simultaneously blocked, the relationship between RT and refusal rate in work trials became closer to that of control monkeys than those treated with $D_2R$ agonists alone, consistent with their impacts on refusal rate (Fig 7B). Therefore, even if $D_1R$ antagonist alone had little effect on workload sensitivity, it may be able to counteract the effect of $D_2R$ treatment under these conditions.

## Discussion

Combining the PET occupancy study and pharmacological manipulation of $D_1R$ and $D_2R$ with quantitative measurement of motivation in monkeys, the current study demonstrated dissociable roles of the DA transmissions via $D_1R$ and $D_2R$ in the computation of the cost–benefits trade-off to guide action. To the best of our knowledge, this is the first study to directly compare the contribution of DA $D_1R$ and $D_2R$ along with the degree of receptor blockade. Using model-based analysis, we showed that DAR blockade had a clear quantitative effect on the sensitivity of animals to information about potential costs and benefits, without any qualitative effect on the way monkeys integrated costs and benefits and adjusted their behavior. We showed that blockade of $D_1R$ or $D_2R$ reduced the incentive impact of reward as the degree of DAR blockade increased, and the incentive impact was more sensitive to the $D_2R$ blockade than the $D_1R$ blockade at lower occupancy. In cost-discounting experiments, we could dissociate the relation between each DAR type and workload versus delay discounting: Workload discounting was increased exclusively by $D_2R$ antagonism, whereas delay discounting was increased by DAR blockade irrespective of receptor subtype. When both $D_1R$ and $D_2R$ were

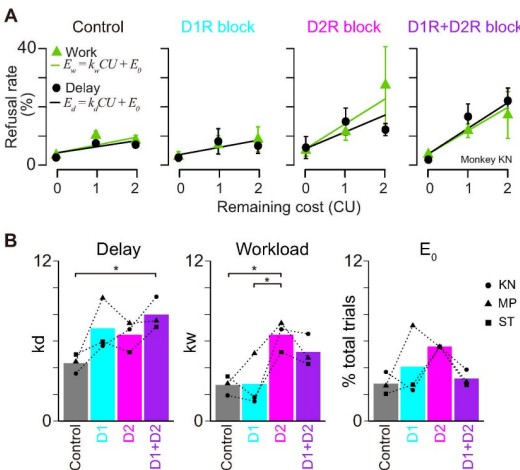

**Fig 7. Effect of both $D_1R$ and $D_2R$ blockades on cost evaluation for motivation. (A)** Representative relationship between refusal rates (in monkey KN; mean ± SEM) and remaining costs for workload (green) and delay trials (black). **(B)** Best-fit parameters, workload discounting ($k_w$), delay discounting ($k_d$), and intercept ($E_0$) are plotted for each treatment condition. Bars and symbols indicate mean and individual data, respectively. $D_1R+D_2R$ indicates the data obtained under both $D_1R$ and $D_2R$ blockades at MO, while $D_1R$ and $D_2R$ blockades at HO resulted in almost no correct performance (see text). All parameters are derived from the best fit of model #1 in S3 Table. Asterisks represent significant difference (*$p < 0.05$, 1-way ANOVA with post hoc Tukey HSD test). The data underlying this figure can be found on the following public repository: https://github.com/minamimoto-lab/2021-Hori-DAR. $D_1R$, $D_1$-like receptor; $D_2R$, $D_2$-like receptor; HO, high occupancy; HSD, honestly significant difference; MO, moderate occupancy.

blocked simultaneously, the effects were synergistic and strengthened for delay discounting, while the effects were antagonistic and diminished for workload discounting. These results suggest that the action of DA is similar between incentive motivation and temporal discounting, but different for workload discounting.

## DA controls the incentive effect of expected reward amount

Previous pharmacological studies have shown that DAR blockade decreased the speed of action and/or probability of engagement behavior [22,23]. However, previous studies did not measure the effect of DAR blockade on incentive motivation in multiple rewarding conditions, and, therefore, data describing the quantitative relationship among DAR stimulation, reward, and motivation are not available. In the present study, we used a behavioral paradigm that enabled us to formulate and quantify the relationship between reward and motivation [2] (Fig 2). Our finding, a reduction of incentive impact due to DAR antagonism (Fig 3), is in line with the incentive salience theory, that is, DA transmission attributes salience to incentive cue to promote goal-directed action [12]. The lack of effect of DA manipulation on satiety and spontaneous water consumption is consistent with previous studies in rodents [57,58]. Our results are also compatible with the idea that DA manipulation mainly influences incentive processes (influence of reward on action) but does not cause a general change of reward processing, which includes hedonic processes (evaluation itself, pleasure associated with consuming reward) [59,60], although further experiments would be necessary to address that point directly [12].

Our model-based analysis indicates that DAR blockade only had a quantitative influence (reduction of incentive impact of reward) without changing the qualitative relationship between reward size and behavior. This is in marked contrast to the reported effects of inactivation of brain areas receiving massive DA inputs, including the orbitofrontal cortex, rostromedial caudate nucleus, and ventral pallidum. Indeed, in experiments using nearly identical tasks and analysis, inactivation or ablation of these regions produced a qualitative change in

the relationship between reward size and behavior (more specifically, a violation of the inverse relationship between reward size and refusal rates) [47,48,61]. Thus, the influence of DAR cannot be understood as a simple permissive or activating effect on target regions. The specificity of the DAR functional role is further supported by the subtle, but significant difference between the behavioral consequences of blocking of $D_1R$ versus $D_2R$. By combining a direct measure of DAR occupancy and quantitative behavioral assessment, the present study demonstrates that the incentive impact of reward is more sensitive to $D_2R$ blockade than $D_1R$ blockade, and, especially, at a lower degree of occupancy (Fig 3C). Moreover, the relationship between occupancy and incentive impact was monotonous for $D_2R$, but tended to be U-shaped for $D_1R$. Although this U-shaped effect of $D_1R$ blockade was inferred solely based on the refusal rate of 2 monkeys without statistical support at the population level and was not found in the RT data, such non-monotonic effects have been repeatedly reported. For example, working memory performance and related neural activity in the prefrontal cortex take the form of an "inverted-U" shaped curve, where too little or too much $D_1R$ activation impairs cognitive performance [62–64]. As for the mechanisms underlying the distinct functional relation between the behavioral effects of $D_1R$ versus $D_2R$ blockade, it is tempting to speculate that this is related to a difference in their distribution, their affinity, and the resulting relation with phasic versus tonic DA action. Indeed, DA affinity for $D_2R$ is approximately 100 times higher than that for $D_1R$ [65]. This is directly in line with the stronger effect of $D_2R$ antagonists at low occupancy levels. Moreover, in the striatum, a basal DA concentration of approximately 5 to 10 nM is sufficient to constantly stimulate $D_2R$. Using available biological data, a recent simulation study showed that the striatal DA concentration produced by the tonic activity of DA neurons (approximately 40 nM) would occupy 75% of $D_2R$ but only 3.5% of $D_1R$ [66]. Thus, blockade of $D_2R$ at low occupancy may interfere with tonic DA signaling, whereas $D_1R$ occupancy would only be related to phasic DA action, i.e., when transient but massive DA release occurs (e.g., in response to critical information about reward). We acknowledge that this remains very hypothetical, but irrespective of the underlying mechanisms, our data clearly support the idea that DA action on $D_1R$ versus $D_2R$ exerts distinct actions on their multiple targets to enhance incentive motivation.

## DA transmission via D1R and $D_2R$ distinctively controls cost-based motivational process

Although many rodent studies have demonstrated that attenuation of DA transmission alters not only benefit but also cost-related decision-making, the exact contribution of $D_1R$ and $D_2R$ remains elusive. For example, reduced willingness to exert physical effort to receive higher reward was similarly found following $D_1R$ and $D_2R$ antagonism in many rodent studies [24,26,59], while it was observed exclusively by $D_2$ antagonism in other studies [21,25]. This inconsistency may have 2 reasons. First, previous studies usually investigated the effect of antagonism on $D_1R$ and $D_2R$ along with a relative pharmacological concentration (e.g., low and high doses). In the present study, PET-assessed DAR manipulation allowed us to directly compare the behavioral effect between $D_1R$ and $D_2R$ with an objective reference, namely occupancy (i.e., approximately 50% and approximately 80% occupancy). Second, the exact nature of the cost (effort versus delay) has sometimes been difficult to identify, and effort manipulation is often strongly correlated with reward manipulation (typically when the amount of reward earned is instrumentally related to the amount of effort exerted; see [10]). Here, using a task manipulating forthcoming workload independently from reward value, we demonstrated that blockade of $D_2R$, but not $D_1R$, increased workload discounting in an occupancy-dependent manner while maintaining linearity (Fig 5F).

Delay discounting and impulsivity—the tendency associated with excessive delay discounting—are also thought to be linked to the DA system [67,68]. Systemic administration of $D_1R$ or $D_2R$ antagonist increases preference for immediate small rewards, rather than larger and delayed rewards [25,32–34]. Concurrently, some of these studies also showed negative effects of $D_1R$ [34] or $D_2R$ blockade [33] on impulsivity. These inconsistencies may be attributed to the differences in behavioral paradigms or drugs (and doses) used. Our PET-assessed DAR manipulation demonstrated that blockade of $D_1R$ and $D_2R$ at the same occupancy level (approximately 50% and approximately 80%) similarly increased delay discounting (Fig 5E), suggesting that DA transmission continuously adjusts delay discounting at the postsynaptic site. This observation is in good accord with the previous finding that increasing DA transmission decreases temporal discounting, e.g., amphetamine or methylphenidate increased the tendency to choose long delays options for larger rewards [32–34,69,70].

In sharp contrast to incentive motivation and delay discounting, which involve both $D_1R$ and $D_2R$, the following 3 observations illustrate a unique mechanism of DA action on workload discounting through $D_2R$ only. First, workload discounting increased only with $D_2R$ antagonism (Fig 5F). Second, the occupancy-dependent effect of $D_2R$ antagonism on the decision–response time relationship was only seen in workload trials (Fig 6). Third, $D_1R$ and $D_2R$ had a synergistic effect in the delay discounting trials, but an antagonistic effect in the workload discounting trials (Fig 7). These results extend previous studies demonstrating increased effort discounting by $D_2R$ blockade [25,71]. Besides, our observation that blocking $D_2R$ increased refusal rates without slowing down responses (Fig 6B) emphasizes the role of DA in effort-based decision-making and supports the notion that DA activation allows overcoming effort costs [21]. This is in apparent contrast to neurophysiological and voltammetry studies that show a lack of sensitivity of DA release to effort but comforts the idea that DA function requires the integration of receptor action on top of neuronal activity and releasing patterns [10,17].

This differential relation between DA and delay versus workload might be related to the differential expression of these receptors in the direct versus indirect striatopallidal pathway, where the striatal neurons exclusively express $D_1R$ and $D_2R$, respectively [72]. Opposing functions between these pathways have been proposed: Activity of the direct pathway ($D_1R$) neurons reflects positive rewarding events promoting movement, whereas activity of the indirect pathway ($D_2R$) neurons is related to negative values mediating aversion or inhibiting movements [56,73,74] (but see [75]). DA increases the excitability of direct pathway neurons, and this effect was reduced by $D_1R$ antagonism, decreasing motor output. DA reduces the responsiveness of indirect pathway neurons via $D_2R$ [72], and blockade of $D_2R$ would increase the activity, reducing motor output via decreased thalamocortical drive [76]. Interestingly, a neural network model has been proposed by considering these opposing DA functions of direct/indirect circuit embedded in reinforcement learning framework, successfully explaining the enhancement of effort cost due to $D_2R$ blockade [77]. This scenario might also explain our finding of a synergistic effect of simultaneous $D_1R$ and $D_2R$ blockade on delay discounting. Further work would be necessary to clarify this hypothesis, including the dynamic relation with tonic versus phasic DA release, but altogether, these data strongly support the concept that distinct neurobiological processes underlie benefits (reward availability) and costs (energy expenditure).

## Limitations of the current study

Finally, limitations of the current study and areas for further research can be discussed. First, there were relatively larger individual disparities in estimated values of $D_2R$ occupancy by

haloperidol in 3 monkeys (Fig 1D), which could reflect individual variance of haloperidol metabolism and/or elimination. However, the time course of the recovery of $D_2R$ occupancy was relatively consistent across subjects, being in line with that of behavioral change. Haloperidol induced long-lasting occupancy for several days, thereby potentially causing unexpected long-term changes, such as synaptic plasticity. Although we cannot eliminate the potential effects of plastic change, a comparable behavioral impact was also observed after raclopride administration, which would induce short-term occupancy (S2 Fig), supporting the view that blockade of DAR reduced motivation in an occupancy-dependent manner. Second, because of applying systemic antagonist administration, the current study could not determine which brain area(s) is responsible for antagonist-induced alterations of benefit- and cost-based motivation. While our data support the notion that differential neural networks involve workload and delay discounting, further study (e.g., local infusion of DA antagonist) is needed to identify the locus of the effects, generalizing our findings to unravel the circuit and molecular mechanism of motivation. We should also note that the current study does not address dynamic learning paradigms, and, therefore, does not generalize our findings to the function of the DA system in learning directly. Despite these limitations, the current study provides unique insights into the role of the DA system in the motivational process.

## Conclusions

In summary, the present study demonstrates a dissociation between the functional role of DA transmission via $D_1R$ and $D_2R$ in benefit- and cost-based motivational processing. DA transmissions via $D_1R$ and $D_2R$ modulate both the incentive impact of reward size and the negative influence of delay. By contrast, workload discounting is regulated exclusively via $D_2R$, since apparently $D_1R$ alone had no role. In addition, $D_1R$ and $D_2R$ had synergistic roles in delay discounting but opposite roles in workload discounting. These dissociations indicate different underlying mechanisms of DA on motivation, which can be attributed to differential involvement of the direct and indirect striatofugal pathways. Together, our findings add an important aspect to our current knowledge concerning the role of DA signaling motivation based on the trade-off between costs and benefits, thus providing an advanced framework for understanding the pathophysiology of psychiatric disorders.

## Materials and methods

### Ethics statement

All surgical and experimental procedures were approved by the Animal Care and Use Committee of the National Institutes for Quantum and Radiological Science and Technology (#09–1035) and were in accordance with the Institute of Laboratory Animal Research Guide for the Care and Use of Laboratory Animals.

### Subjects

A total of 9 male adult macaque monkeys (8 Rhesus and 1 Japanese; 4.6 to 7.7 kg) were used in this study. All monkeys were individually housed. Food was available ad libitum, and motivation was controlled by restricting access to fluid to experimental sessions, when water was delivered as a reward for performing the task. Animals received water supplementation whenever necessary (e.g., if they could not obtain enough water during experiments), and they had free access to water whenever testing was interrupted for more than a week. For environmental enrichment, play objects and/or small foods (fruits, nuts, and vegetables) were provided daily in the home cage.

## Drug treatment

All experiments in this study were carried out with injected i.m. SCH23390 (Sigma-Aldrich, St. Louis, MO), haloperidol (Dainippon Sumitomo Pharma, Japan), and raclopride (Sigma-Aldrich) dissolved or diluted in 0.9% saline solution. Animals were pretreated with an injection of SCH23390 (10, 30, 50, or 100 μg/kg), haloperidol (10 μg/kg), or raclopride (10 or 30 μg/kg) 15 minutes before the beginning of behavioral testing or PET scan. In behavioral testing, saline was injected as a vehicle control by the same procedure as the drug treatment. The administered volume was 1 mL across all experiments with each monkey.

## Surgery

Four monkeys underwent surgery to implant a head-hold device for the PET study using aseptic techniques [78]. We monitored body temperature, heart rate, $SpO_2$, and tidal $CO_2$ throughout all surgical procedures. Monkeys were immobilized by i.m. injection of ketamine (5 to 10 mg per kg) and xylazine (0.2 to 0.5 mg per kg) and intubated with an endotracheal tube. Anesthesia was maintained with isoflurane (1% to 3%, to effect). The head-hold device was secured with plastic screws and dental cement over the skull. After surgery, prophylactic antibiotics and analgesics were administered. The monkeys were habituated to sit in a primate chair with their heads fixed for approximately 30 minutes for more than 2 weeks.

## PET procedure and occupancy measurement

Four monkeys were used in the measurement. PET measurements were performed with 2 PET ligands: [11C]SCH23390 (for studying $D_1R$ binding) and [11C]raclopride (for studying $D_2R$ binding). The injected radioactivities of [11C]SCH23390 and [11C]raclopride were 91.7 ± 6.0 MBq (mean ± SD) and 87.0 ± 4.9 MBq, respectively. Specific radioactivities of [11C]SCH23390 and [11C]raclopride at the time of injection were 86.2 ± 40.6 GBq/μmol and 138.2 ± 70.1 GBq/μmol, respectively. All PET scans were performed using an SHR-7700 PET scanner (Hamamatsu Photonics, Japan) under conscious conditions and seated in a chair. After transmission scans for attenuation correction using a $^{68}Ge$–$^{68}Ga$ source, a dynamic scan in three-dimensional (3D) acquisition mode was performed for 60 minutes ([11C]SCH23390) or 90 minutes ([11C]raclopride). The ligands were injected via crural vein as a single bolus at start of the scan. All emission data were reconstructed with a 4.0-mm Colsher filter. Tissue radioactive concentrations were obtained from VOIs placed on several brain regions where DARs are relatively abundant: caudate nucleus, putamen, nucleus accumbens (NAcc), thalamus, hippocampus, amygdala, parietal cortex, principal sulcus (PS), dorsolateral prefrontal cortex (dlPFC), and ventrolateral prefrontal cortex (vlPFC), as well as the cerebellum (as reference region). Each VOI was defined on individual T1-weighted axial magnetic resonance (MR) images (EXCELART/VG Pianissimo at 1.0 tesla, Toshiba, Japan) that were co-registered with PET images using PMOD image analysis software (PMOD Technologies, Switzerland). Regional radioactivity of each VOI was calculated for each frame and plotted against time. Regional binding potentials relative to non-displaceable radioligands ($BP_{ND}$) of $D_1R$ and $D_2R$ were estimated with a simplified reference tissue model on VOI and voxel-by-voxel bases [79–81]. The monkeys were scanned with and without drug treatment condition on different days.

Occupancy levels were determined from the degree of reduction (%) of $BP_{ND}$ by antagonists [82]. DAR occupancy was estimated as follows:

$$Occupancy(\%) = (1 - BP_{NDTreament}/BP_{NDBaseline}) \times 100 \qquad (3)$$

where $BP_{ND\ Baseline}$ and $BP_{ND\ Treatment}$ are $BP_{ND}$ measured without (baseline) and with an

antagonist, respectively. Relationship between $D_1R$ occupancy ($D_1Occ$) and dose of SCH23390 (Dose) was estimated with 50% effective dose ($ED_{50}$) as follows:

$$D_1Occ(\%) = 100 \times Dose/(ED50 + Dose) \tag{4}$$

Relationship between $D_2R$ occupancy ($D_2Occ$) and days after haloperidol injection was estimated using the level at day 0 with a decay constant ($\lambda$) as follows:

$$D_2Occ(\%) = Occ_{Day0}e^{-\lambda Day} \tag{5}$$

## Behavioral tasks and testing procedures

Three monkeys (ST, 6.4 kg; KN, 6.3 kg; MP, 7.3 kg) were used for the behavioral study. For all behavioral training and testing, each monkey sat in a primate chair inside a sound-attenuated dark room. Visual stimuli were presented on a computer video monitor in front of the monkey. Behavioral control and data acquisition were performed using the REX program. Neurobehavioral Systems Presentation software was used to display visual stimuli (Neurobehavioral Systems, www.neurobs.com). We used 2 types of behavioral tasks, reward-size task and work/delay task, as described previously [2,51]. Both tasks consisted of color discrimination trials (see Figs 2A and 5A). Each trial began when the monkey touched a bar mounted at the front of the chair. The monkey was required to release the bar between 200 and 1,000 ms after a red spot (wait signal) turned green (go signal). In correctly performed trials, the spot then turned blue (correct signal). A visual cue was presented at the beginning of each color discrimination trial (500 ms before the red spot appears).

In the reward-size task, a reward of 1, 2, 4, or 8 drops of water (1 drop = approximately 0.1 mL) was delivered immediately after the blue signal. Each reward size was selected randomly with equal probability. The visual cue presented at the beginning of the trial indicated the number of drops for reward (Fig 2A).

In the work/delay task, a water reward (approximately 0.25 mL) was delivered after each correct signal immediately or after an additional 1 or 2 instrumental trials (work trial) or after a delay period (delay trials). The visual cue indicated the combination of the trial type and requirement to obtain a reward (Fig 5A). Pattern cues indicated the delay trials with the timing of reward delivery after a correct performance: either immediately (0.3 seconds, 0.2 to 0.4 seconds; mean, range), short delay (3.6 seconds, 3.0 to 4.2 seconds), or long delay (7.2 seconds, 6.0 to 8.4 seconds). Gray scale cues indicated work trials with the number of trials the monkey would have to perform to obtain a reward. We set the delay durations to be equivalent to the duration for 1 or 2 color discrimination trials, so that we could directly compare the cost of 1 or 2 arbitrary units (cost unit; CU).

If the monkey released the bar before the green target appeared or within 200 ms after the green target appeared or failed to respond within 1 second after the green target appeared, we regarded the trial as a "refusal trial"; all visual stimuli disappeared, the trial was terminated immediately, and after the 1-second intertrial interval, the trial was repeated. Our behavioral measurement for the motivational value of outcome was the proportion of refusal trials. Before each testing session, the monkeys were subject to approximately 22 hours of water restriction in their home cage. Each session continued until the monkey would no longer initiate a new trial (usually less than 100 minutes).

Before this experiment, all monkeys had been trained to perform color discrimination trials in cued multi-trial reward schedule task [43] for more than 3 months. The monkeys were tested with the reward-size task and work/delay task for more than 2 months as training to become familiar with the cueing condition.

Each monkey was tested from Monday to Friday. Treatment with SCH23390 was performed every 4 or 5 days. On other days without SCH23390, sessions with saline (1 mL) treatment were analyzed as control sessions. Haloperidol was given every 2 or 3 weeks on Monday or Tuesday, because $D_2R$ occupancy persisted for several days after a single dose of haloperidol treatment (Fig 1D). The days before haloperidol treatment were analyzed as control sessions. Each dose of SCH23390 or a single dose of haloperidol was tested 4 or 5 times with the reward-size task and at least 3 times with the work/delay task per each animal.

## Sucrose preference test

Two monkeys (RO, 5.8kg; KY, 5.6kg) were used for the sucrose preference test. The test was performed in their home cages once a week. In advance of the test, water access was prevented for 22 hours. The monkeys were injected with SCH23390 (30 μg/kg), haloperidol (10 μg/kg), or saline 15 minutes before the sucrose preference test. Two bottles containing either 1.5% sucrose solution or tap water were set into bottle holders in the home cage, and the monkeys were allowed to freely consume fluids for 2 hours. The total amount of sucrose (SW) and tap water (TW) intake was measured and calculated as sucrose preference index (SP) as follows: SP = (SW–TW) / (SW + TW). The position of sucrose and tap water bottles (right or left toward the front panel of the home cage) was counterbalanced across sessions and monkeys. Drugs or saline was injected alternatively once a week. We also measured the osmolality level in blood samples (1 mL) obtained immediately before and after each testing session.

## Behavioral data analysis

All data and statistical analyses were performed using the R statistical computing environment (www.r-project.org). The average error rate for each trial type was calculated for each daily session, with the error rates in each trial type being defined as the number of error trials divided by the total number of trials of that given type. The monkeys sometimes made many errors at the beginning of the daily session, probably due to high motivation/impatience; we excluded the data until the first successful trial in these cases. A trial was considered an error trial if the monkey released the bar either before or within 200 ms after the appearance of the green target (early release) or failed to respond within 1 second after the green target (late release). We did not distinguish between the 2 types of errors and used their sum except for the error pattern analysis. We performed repeated-measures ANOVAs to test the effect of treatment × reward size (for data in reward-size task) on RT, on late release rate (i.e., error pattern). Post hoc comparisons were performed using Tukey HSD test, and a priori statistical significance was set at = 0.05.

We used the refusal rates to estimate the level of motivation because the refusal rates of these tasks (E) are inversely related to the value for action [2]. In the reward-size task, we used the inverse function (Eq 1). We fitted the data to LMMs [83], in which the random effects across DAR blockade conditions on parameter a and/or intercept e (Fig 2C) were nested. Model selection was based on Bayesian information criterion (BIC), an estimator of in-sample prediction error for the nested models (S1 Table). Using the selected model, the parameter a was estimated individually and then normalized by the value in nontreated condition (control, CON) (Fig 3A and 3B).

In the work/delay task, we used linear models to estimate the effect of remaining cost, i.e., workloads and delay, as described previously [55],

$$E_w = k_w CU + E_0 \tag{6}$$

$$E_d = k_d CU + E_0 \tag{7}$$

where $E_w$ and $E_d$ are the error rates, and $k_w$ and $k_d$ are workload discounting and delay discounting parameters, respectively. $CU$ is the number of remaining CUs, and $E_0$ is the intercept. We used LMMs to estimate the effect of DAR blockade on the discounting parameters. We imposed the constraint that the intercept ($E_0$) has the same value across trials and assumed the base statistical model in which the random effects of the 2 receptor types (delay and workload) affect the regression confidents independently. Four models were nested to consider the presence or absence of random effects, random effects of treatment conditions, and subjects (S3 Table). The best model was selected based on BIC for the entire data set, which is the sum of the regression results for each unit faceted by individual and/or treatment condition. For example, model #1 was fit to a total of 18 data sets (3 monkeys × 3 treatment conditions (CON, MO, and HO) × 2 subtypes ($D_1R$ and $D_2R$), and then BIC was calculated by the sum of each fitting. Modeling was performed with the lme4 package in R, and the parameters (e.g., $k_w$ and $k_d$) were estimated from the model. We performed 1-way ANOVAs to test the significance of the effect of treatment on discounting parameters with post hoc Tukey HSD test.

LMMs were also applied for the correlation analysis between refusal rate ($E$) and reaction time ($Rt$) (Figs 4 and 6 and S5), where 4 statistical models were nested to take into account the presence or absence of random effects of subjects and treatment conditions, and the best-fit model was selected based on BIC (S2, S4 and S5 Tables).

## Supporting information

**S1 Table. Model comparison the effect of DAR blockade on refusal rates in reward-size task (for Fig 2).** *a(cond)* and *e(cond)* indicate the random effects of DAR blocking treatment conditions on parameters *a* and *e*, respectively. BIC is a relative measure of quality for the models (#1–4). ΔBIC denotes difference from minimum BIC. BIC, Bayesian information criterion; DAR, DA receptor.
(PDF)

**S2 Table. Model comparison for the effect of DAR blockade on the relationship between refusal rate and RT in reward-size task (for Fig 4).** *(Rt|\*)* indicates random effects on regression parameters. *E*, refusal rate; *Rt*, reaction time; *cond*, treatment condition; *monkey*, subject. DAR, DA receptor.
(PDF)

**S3 Table. Model comparison for the effect of DAR blockade on refusal rates in work/delay task (for Fig 5).** *CU* and $E_0$ indicate remaining cost and intercept, respectively. *(0+ CU|\*)* and *(CU|\*)* indicate random effects on both regression coefficient and intercept ($E_0$) or on regression coefficient alone, respectively. *E*, refusal rate; *type*, trial type (delay or work); *cond*, treatment condition (CON, MO, and HO for $D_1R$ and $D_2R$ blocking); *monkey*, subject. CON, control; $D_1R$, $D_1$-like receptor; $D_2R$, $D_2$-like receptor; DAR, DA receptor; HO, high occupancy; MO, moderate occupancy.
(PDF)

**S4 Table. Model comparison for the effect of DAR blockade on the relationship between refusal rate and RT in work/delay task (for Fig 6).** *(Rt|\*)* indicates random effects on regression parameters. *E*, refusal rate; *Rt*, reaction time; *cond*, treatment condition; *monkey*, subject. DAR, DA receptor.
(PDF)

**S5 Table. Model comparison for the effect of both $D_1R$ and $D_2R$ blockades on the relationship between refusal rate and RT in work/delay task (for S5 Fig).** *(Rt|\*)* indicates random

effects on regression coefficient. *E*, refusal rate; *Rt*, reaction time; *cond*, treatment condition; *monkey*, subject. $D_1R$, $D_1$-like receptor; $D_2R$, $D_2$-like receptor.
(PDF)

**S1 Fig. Occupancy estimation.** Example of occupancy estimation based on modified Lassen plot of [$^{11}$C]SCH23390 PET data obtained from monkey DO. Colored dots represent the relationship between decreased specific binding [i.e., $BP_{ND}$ (baseline)–$BP_{ND}$ (blocking)] and baseline [$BP_{ND}$ (baseline)] for each brain region under each blocking condition (indexed by color). Occupancy was determined as a proportion of reduced specific binding to baseline, which corresponds to the slope of linear regression. In this case, $D_1$ occupancy was 80%, 78%, 67%, and 26% for 100, 50, 30, and 10 µg/kg doses, respectively. The data underlying this figure can be found on the following public repository: https://github.com/minamimoto-lab/2021-Hori-DAR. PET, positron emission tomography.
(PDF)

**S2 Fig. Comparable effects of $D_2R$ antagonism between raclopride and haloperidol at similar occupancy. (A)** Occupancy of $D_2R$ measured at striatal ROI is plotted against dose of raclopride. **(B)** Error rates as a function of reward size for control (black) and after injection of raclopride (10 µg/kg, i.m. left side) and haloperidol (10 µg/kg, i.m. right side) in monkey KN are plotted. Dotted curves are best-fit inverse function (model #1 in S1 Table). The data underlying this figure can be found on the following public repository: https://github.com/minamimoto-lab/2021-Hori-DAR. $D_2R$, $D_2$-like receptor; ROI, region of interest.
(PDF)

**S3 Fig. Effect of $D_1R/D_2R$ blockade on RT and error pattern. (A, B)** Cumulative distribution of RT for control and $D_1R$ blockade conditions in drop-1 and drop-8 trials, respectively. **(C)** Mean RT as function of reward size for control and $D_1R$ blockade conditions. Two-way ANOVA, reward × condition; main effect of condition, $F_{(4, 164)} = 109.8$, $p < 1.0 \times 10^{-15}$; main effect of reward, $F_{(3, 164)} = 111.0$, $p < 10^{-15}$; interaction, $F_{(12, 164)} = 4.7$, $p < 1.0 \times 10^{-5}$. **(D)** Late release rate (mean ± SEM) as function of reward size for control and $D_1R$ blockade conditions. Two-way ANOVA, reward × condition; main effect of condition, $F_{(4, 163)} = 18.6$, $p < 1.0 \times 10^{-11}$; main effect of reward, $F_{(3, 163)} = 9.8$, $p < 10^{-5}$; interaction, $F_{(12, 163)} = 1.0$, $p = 0.4$. **(E–H)** Same as (A–D), but for $D_2R$ blockade. RT; main effect of condition, $F_{(6, 92)} = 7.2$, $p < 1.0 \times 10^{-5}$; main effect of reward, $F_{(3, 92)} = 81.9$, $p < 10^{-15}$; interaction, $F_{(18, 164)} = 0.6$, $p = 0.65$. Late release rate; main effect of condition, $F_{(6, 90)} = 3.5$, $p = 0.0038$; main effect of reward, $F_{(3, 90)} = 19.2$, $p < 10^{-9}$; interaction, $F_{(18, 90)} = 1.4$, $p = 0.14$. * significantly different from control, $p < 0.05$ post hoc Tukey HSD. Data were obtained from monkey ST. The data underlying this figure can be found on the following public repository: https://github.com/minamimoto-lab/2021-Hori-DAR. $D_1R$, $D_1$-like receptor; $D_2R$, $D_2$-like receptor; HSD, honestly significant difference; RT, reaction time.
(PDF)

**S4 Fig. Little influence of DAR blockade on sucrose preference and blood osmolality. (A)** Sucrose preference index after administration of saline (Control), SCH23390 (30µg/kg, $D_1$), and haloperidol (10µg/kg; $D_2$, day 0), respectively. There was no significant effect of DAR blockade on overall intake (1-way ANOVA, treatment, monkey KY, $F_{(2, 8)} = 1.26$, $p = 0.33$; monkey RO, $F_{(2, 14)} = 2.01$, $p = 0.17$) or sucrose preference (1-way ANOVA; treatment, monkey KY, $F_{(2, 8)} = 1.62$, $p = 0.26$; monkey RO, $F_{(2, 8)} = 1.38$, $p = 0.31$). **(B)** Blood osmolality measured in serum samples obtained before (pre) and after (post) sucrose test. There was no significant impact of DAR blockade (2-way ANOVA, monkey KY, main effect of treatment, $F_{(2, 10)} = 4.0$, $p = 0.056$; pre-post, $F_{(1, 10)} = 93.83$, $p = 2.1 \times 10^{-6}$, interaction, $F_{(2,10)} = 0.74$,

$p = 0.50$; monkey RO, treatment, $F_{(2, 20)} = 1.22$, $p = 0.32$; pre-post, $F_{(1, 20)} = 40.8$, $p = 3.1 \times 10^{-6}$, interaction, $F_{(2,20)} = 0.13$, $p = 0.88$). Filled circles and shades indicate median and raw data points, while horizontal bars indicate SD. The data underlying this figure can be found on the following public repository: https://github.com/minamimoto-lab/2021-Hori-DAR. DAR, DA receptor.
(PDF)

**S5 Fig. Effect of both $D_1R$ and $D_2R$ blockades on the relationship between refusal rate and RT. (A)** Relationship between refusal rate and average RT for each reward size in session by session for $D_2$ blocking and $D_1+D_2$ blocking in delay trials. Data are plotted individually for monkeys KN, MP, and ST, in order from top to bottom. Colors indicate treatment condition. Thick lines indicate linear regression lines (model #1 in S5 Table). **(B)** Same as A, but for workload trials. Note that for the data in workload trials, a multiple linear model with random effect of condition (model #3 in S5 Table) was chosen as the best model to explain the data, where the steepness of the slope under $D_1+D_2$ treatment was the same as that of control. The data underlying this figure can be found on the following public repository: https://github.com/minamimoto-lab/2021-Hori-DAR. $D_1R$, $D_1$-like receptor; $D_2R$, $D_2$-like receptor; RT, reaction time.
(PDF)

## Acknowledgments

We thank R. Suma, T. Okauchi, Y. Sugii, R. Yamaguchi, Y. Matsuda, and J. Kamei for their technical assistance, and K. Oyama for discussion. We also thank Dr. M-R. Zhang and his colleagues at the Department of Radiopharmaceuticals Development, QST, for producing the radioligands. A Japanese monkey used in this study was provided by National Bio-Resource Project "Japanese Monkeys" of MEXT, Japan.

## Author Contributions

**Conceptualization:** Takafumi Minamimoto.

**Formal analysis:** Yukiko Hori, Yuji Nagai, Koki Mimura, Takafumi Minamimoto.

**Funding acquisition:** Tetsuya Suhara, Makoto Higuchi, Takafumi Minamimoto.

**Investigation:** Yukiko Hori, Yuji Nagai.

**Project administration:** Takafumi Minamimoto.

**Supervision:** Tetsuya Suhara, Makoto Higuchi, Takafumi Minamimoto.

**Visualization:** Yukiko Hori, Yuji Nagai, Koki Mimura, Takafumi Minamimoto.

**Writing – original draft:** Yukiko Hori, Takafumi Minamimoto.

**Writing – review & editing:** Yukiko Hori, Yuji Nagai, Koki Mimura, Tetsuya Suhara, Makoto Higuchi, Sebastien Bouret, Takafumi Minamimoto.

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
