## [Editor Report · Decision Letter 0]

10 Nov 2020

Dear Dr Minamimoto, 

Thank you for submitting your manuscript entitled "Differential Contribution of Dopaminergic Transmission at D1- and D2-like Receptors to Cost/Benefit Evaluation for Motivation in Monkeys" for consideration as a Research Article by PLOS Biology.

Your manuscript has now been evaluated by the PLOS Biology editorial staff as well as by an academic editor with relevant expertise and I am writing to let you know that we would like to send your submission out for external peer review.

Please re-submit your manuscript within two working days, i.e. by Nov 12 2020 11:59PM.

Given the disruptions resulting from the ongoing COVID-19 pandemic, please expect delays in the editorial process. We apologize in advance for any inconvenience caused and will do our best to minimize impact as far as possible. 

Kind regards,

Lucas Smith, Ph.D.,

Associate Editor

PLOS Biology

---

## [Decision Letter · Decision Letter 1]

16 Dec 2020

Dear Dr Minamimoto,

Thank you very much for submitting your manuscript "Differential Contribution of Dopaminergic Transmission at D1- and D2-like Receptors to Cost/Benefit Evaluation for Motivation in Monkeys" for consideration as a Research Article at PLOS Biology. Your manuscript has been evaluated by the PLOS Biology editors, an Academic Editor with relevant expertise, and by several independent reviewers.

As you will see from their detailed responses (below), all of the reviewers are enthusiastic about the overall potential of this study. However, the reviewers do ask that you provide some additional clarifications of your ideas, and provide a better grounding of these findings and mechanistic suggestions within the broader literature across species. Additionally, Reviewer 2 raises some technical concerns, particularly around the ANOVA analyses, D2 occupancy estimates, and ability to rule out a motor impairment.

In light of the reviews, we will not be able to accept the current version of the manuscript, but we would welcome re-submission of a much-revised version that takes into account the reviewers' comments. We cannot make any decision about publication until we have seen the revised manuscript and your response to the reviewers' comments. Your revised manuscript is also likely to be sent for further evaluation by the reviewers.

We expect to receive your revised manuscript within 3 months. 

**IMPORTANT - SUBMITTING YOUR REVISION**

*Re-submission Checklist*

*Published Peer Review*

*PLOS Data Policy*

*Blot and Gel Data Policy*

Sincerely,

Lucas Smith, Ph.D.,

Associate Editor,

lsmith@plos.org,

PLOS Biology

REVIEWS:

Reviewer's Responses to Questions

PLOS authors have the option to publish the peer review history of their article (what does this mean?). If published, this will include your full peer review and any attached files.

Reviewer #1: No

Reviewer #2: No

Reviewer #3: Yes: John D Salamone

Reviewer #1: This is the first study to directly compare the contribution of the dopamine D1R and D2R by showing the degree of biding to the striatum and other brain structures. I think the series of results that the authors presented are very convincing and the experimental procedure of the authors is straightforward and excellent. I have a few following concerns. If the authors revise the manuscript considering the following comments, the paper will be much improved. The task paradigm and the obtained change in behaviors are very convincing because of the established procedure of the author's series of experiments. However, for the readers who are not familiar with the task procedures, it is better to add a detailed explanation of the task and add further interpretation of the results. For example, each assignment and the experimental process is accurately described, but the conclusions are drawn from them, and their interpretation is not emphasized. As a result, each assignment's significance and its importance to the reader is often not conveyed to the reader. 

Major comments:

(1) Regarding the data in Figure 1, it is very important to examine the degree of D1R and D2R block by examining the occupancy with PET, and I think it is clean data. However, I fear that long-term plastic changes may be a side effect, especially in D2R, because of the long-lasting occupancy. The possibility of mixed plastic changes caused by long-term blocks needs to be discussed in detail.

(2) On the relationship between reaction time and refusal rate, at L201, the authors mentioned, "D1R blockade also influences response speed, probably due to slowing cognitive processing". If the motivation itself is decreased by D1R block, it may directly affect reaction time and refusal rate. In this case, it is not necessarily related to whether there is some disorder to cognitive processing or not. The interpretation of this behavior, especially the relationship between RT and the refusal rate, needs to be discussed in detail.

(3) "Relative reward" in L213-215 is an abrupt concept that needs to be explained; what is the purpose of the Sucrose preference test?

(4) About L300-302, the authors introduced concepts of "delay discounting" and "workload discounting." Although the relationship between them was carefully discussed in the Discussion, I would appreciate it if the authors add detail conceptual details when they introduce them (i.e., at Results). At least, the author should add "why" they focused on these parameters. 

(5) About the relationship between delay and workload discounting, I have naïve questions. First, can we directly compare the two in a paradigm? And how are those two concepts distinguished in monkeys?

(6) The circuitry changes due to the D1R and D2R blocks are discussed in the Discussion. Of course, there are many difficulties with circuit changes, but is there anything added from the changes seen in PET that we can add to the Discussion?

Reviewer #2: The authors present a pharmacological study addressing the roles of D1 and D2 receptor stimulation on how monkeys process reward incentives and discount effort or waiting time while they do variations of a task to collect rewards. The authors tested the animals either after receiving saline or different doses of SCH23390 to test the involvement of D1 receptors. For the D2 receptors, they administered a fixed dose of haloperidol and tested the animals on different days over a period of 8 days. This was done instead of administering different doses because haloperidol has a high half-life and stays long enough in the system to be able to test the animals with different effective doses. 

What appears to be the key feature of the study is that the authors used PET to quantify the receptor occupancy achieved by the doses used in the behavioral experiments. Furthermore, they tested the behavioral impact of different levels of occupancy, effectively doing a dose/response curve for blockade of D1 and D2 receptors. 

These are definitely valuable data on a hot topic in systems neurosciences. The question of the relative contribution of D1 and D2 receptors to the dopamine modulation of incentive processing and motivation is timely, and I think that the paper has potential to contribute to clarifying it. However, the manuscript in its current form is far from satisfactory. The presentation is rather poor and requires lots of suppositions from the reader. Thus, there are some aspects when I am not sure if the current analyses are well grounded. Yet in some instances, the authors make unwarranted claims. 

- Presentation - even though the details of the methods are given after the results section, it should be possible to know what the authors did while reading the results section. Details such as the number of monkeys for each experiment should be stated straight away before giving any specific results. Similarly, the authors should state: that the monkeys are not the same for the PET and the pharmacological experiments; the gist of the PET analysis (at least name the method used for analysis; whether the analysis is ROI or voxel-based, which ROIs do the results refer to); whether the quantification of receptor occupancy for SCH23390 was done in different sessions; how many (or range of) months of training; the models that are tested to ultimately infer the parameter a in the first experiment should be spelled out so that it is possible to know what the authors are testing before looking at the supplement; define what is meant by moderate dose and high dose in the workload and time discounting study and how these doses were derived; and that the reward size in the discounting study is constant. 

- It was unclear how the ANOVAs were performed and how the data from the two or three individual monkeys are treated. Given that there are only 2 or 3 monkeys and that it is impossible to make any random effects inference, I would suggest that the authors analyze each individual separately. This is the most honest way to present the results. Conclusions about the effects of the drugs on monkeys in general is not possible, but this does not mean that this data is not very valuable as human studies cannot provide the value occupancy information and test the range of doses. However, the data needs to be interpreted rightly. Furthermore, ANOVAs need to be followed up with post hoc tests in order to be able to interpret interactions.

- Related to the previous point, I do not believe that the authors can say that the effects of D1 antagonism are quadratic when they have evidence for a quadratic effect in one monkey and evidence for saturating monotonic effects in the other monkey. I would only dear to make inferences when the effects are consistent across tested monkeys. And even in this case, inference will need to be tentative as far as random effects cannot be tested. Similarly, the authors overinterpret the mean effects of combining antagonists as the effects do not seem to be consistent across monkeys.

- Whereas the estimates of occupancy based on administered dose of SCH23390 appears to be very robust, the estimates of occupancy for haloperidol is much more uncertain, provably reflecting individual differences in the metabolism and elimination of haloperidol. This is important given that occupancy was not measured for the animals that took part in the behavioral experiments. This limitation should be acknowledged and discussed.

- At multiple points in the manuscript the authors present their ideas and results without enough consideration of previous evidence. Some examples: the authors omit a paper by Gao et al in Nature Neuroscience providing evidence that dopamine neurons are not sensitive to effort costs; Husain's lab in Oxford have some nice studies quantifying the effects of effort discounting and its modulation by dopamine; Berke's lab in Seattle have provided very important insights on the differential roles of dopamine in learning and performance.

- What do the authors mean by " However, the previous studies did not address the quantitative effect of DAR blockade on incentive motivation; more specifically, there was a lack of experimental data to model the causal relationship among DAR stimulation, reward, and motivation." Certainly, there are previous studies showing that D1 and D2 blockade affects incentive motivation.

- Michael Frank models of the direct and indirect pathway suggest a division of labor between D1 and D2 receptor whereby D1 and D2 affect learning from positive and negative reward prediction errors, respectively, but also have different impacts on choice performance and motivation. The latter being strongly controlled by D2 receptors. See, for instance, the excellent review by Collins and Frank in Psychological Reviews. The authors should discuss this model and how it plays on their results.

- on page 12, how can you rule out that the effects are not related to cognitive but motor impairment? Related to that, in the sentence, "Thus, the effect of D2 manipulation on value-based decision was relatively independent from the effects on cognitive or motor speed itself" is not appropriate. Motor speed is what is measured in the experiment, cognition or motor impairments are hypothesized mechanisms. They should not be mixed up and equated in one and the same sentence. 

- On page 29, the authors state that 3 monkeys, whose codes are provided, were used for the behavioral experiments. This seem to apply to the discounting experiment but according to the results section and figures, 2 different monkeys were used for the incentive experiment. 

- In figure 1, there seems to be missing some data points. Does it imply that not all doses were tested in all subjects? 

- In the introduction, for the sentence "Blockade of either D1R or D2R biases animals' choices in tasks manipulating the cost/benefits trade-off. These biases should be spelled out. 

- Why was a head hold device implanted?

Reviewer #3: Overall, this was an interesting and potentially very useful manuscript for the field, focusing on the effects of DA antagonists on several distinct aspects of motivation. The behavioral paradigms are elegant, and the use of monkeys is critical, because most of the animal studies in this area have been with rodents, and monkeys potentially provide a bridge between the rodent and human literatures. Moreover, the receptor occupancy data are a very important addition. As one can see from my comments, this manuscript stimulated a lot of thinking. My specific comments are listed below.

Comments:

In the first paragraph of the introduction, the authors state "For motivational value computation, the expected value of benefits (i.e., rewards) has a positive influence, while the cost necessary to earn the expected reward has a negative impact and discounts the net value of reward". Although one sees wording like this in the literature, it is actually misleading. As written, this statement implies that the reward itself is devalued (i.e., "…net value of reward"). However, when considering what is involved in cost/benefit analyses such as this, the net value is actually the relative value of the whole complex activity related to the reinforcer itself plus the instrumental response with all its associated costs. This issue also was evident in the literature on response/reinforcement matching, going back to the 1960s and 70s. While a reinforcement value parameter was determined from response/reinforcement matching studies of variable interval responding, it was evident to some researchers that this parameter was not the value of the reward per se, but rather, the value of the whole activity involving the instrumental response and the delivery of the reinforcer (i.e., the entire sequence of behavior that includes the response, its associated biases, and the reinforcer; Williams 1988). Indeed, the focus of the present article on the lack of effect of DA antagonism on relative reward value highlights the importance of these subtle distinctions in wording and terminology. 

On a related note, in rodent studies it has been shown that in parallel experiments that involve intake of and preference for the reinforcers used in costs/benefit procedures, DAergic manipulations did not affect reinforcer intake or preference (Nunes et al. 2013; Yang et al. 2020). Moreover, studies involving the progressive ratio/chow feeding choice task in rats, which essentially functions as a discounting task involving increasing ratio requirements, it has been shown that D1 and D2 receptor antagonism produce effects that are very different from the effect produced by reinforcer devaluation (Randall et al. 2012, 2014). These findings are consistent with the conclusion of the authors that DA receptor blockade did not produce a general alteration of reinforcement value. 

There have been a large number of rodent studies showing that DA D1 antagonism altered decision making based upon physical effort (Cousins et al. 1994; Nowend et al. 2001; Salamone et al. 2002; Sink et al. 2008; Randall et al. 2014; Yohn et al. 2015; Hosking et al. 2015). The authors seem to suggest that these effects may be solely due to actions on incentive motivation, rather than something to do with work load per se. However, I wonder if there is a possibility that there are species differences; i.e., perhaps the role of direct vs. indirect pathways and hence the effects of D1 antagonism are different in rodents vs. primates. Also, the authors should clarify if the implication of their line of thinking is that blunting of incentive motivation leads to an apparent lack of exertion of effort due to a reduced propensity for action, or if something else is going on. 

One of the implications of these studies, taken as a whole, is the incentive motivation, exertion of effort, delay discounting, and reward value are dissociable from each other. This is consistent with much of the published literature, and is touched upon in the discussion, but is worth emphasizing more because it is an important point. In addition, I would like to see a bit more emphasis on what the authors mean be incentive motivation, in terms of behavior theory.

 Minor Comments:

P 4, "DAR blockades" should be "DAR blockade"

P 16, "following D1 and D2 antagonisms" should be "following D1 and D2 antagonism"

---

## [Decision Letter · Decision Letter 2]

18 May 2021

Dear Dr Minamimoto,

Thank you for submitting your revised Research Article entitled "Differential Contribution of Dopaminergic Transmission at D1- and D2-like Receptors to Cost/Benefit Evaluation for Motivation in Monkeys" for publication in PLOS Biology. I have now obtained advice from the original reviewers and have discussed their comments with the Academic Editor. 

The reviews of your manuscript are appended below. As you will see, all three reviewers feel that you have satisfactorily addressed their previous comments. However reviewer 2 has noted a few lingering minor issues with the manuscript which we think can be addressed via textual changes.

Based on the reviews, we will probably accept this manuscript for publication, provided you satisfactorily address the remaining points raised by the reviewers. **IMPORTANT: Please also make sure to address the following data and other policy-related requests outlined here:

1) ETHICS REQUEST: Please provide information about the housing and environmental enrichment used for your monkeys, including whether they were group housed. For experiments requiring surgery, please describe the steps taken to minimize suffering, including use of anesthesia.

2) DATA REQUEST: Thank you for providing the data underlying each figure. Please add a sentence to each figure legend (including supplementary figures), noting where the underlying data can be found. For example, you could say "the data underlying this figure can be found on the following public repository: https://github.com/minamimoto-lab/2021-Hori-DAR.”

3) I have discussed the title of your manuscript with my colleagues, and we wonder if it might be streamlined a bit to make it more accessible. For example, we suggest the following title: "D1- and D2-like Receptors differentially mediate the effects of dopaminergic transmission on Cost/Benefit Evaluation and Motivation in Monkeys"

4) BLURB: Please provide a blurb which (if accepted) will be included in our weekly and monthly Electronic Table of Contents, sent out to readers of PLOS Biology, and may be used to promote your article in social media. The blurb should be about 30-40 words long and is subject to editorial changes. It should, without exaggeration, entice people to read your manuscript. It should not be redundant with the title and should not contain acronyms or abbreviations. For examples, view our author guidelines: https://journals.plos.org/plosbiology/s/revising-your-manuscript#loc-blurb

5) I noticed a typo in figure 5 (there are two panels labeled 5D). 

We expect to receive your revised manuscript within two weeks. 

*Published Peer Review History*

*Early Version*

Sincerely,

Lucas Smith, Ph.D.,

Associate Editor,

lsmith@plos.org,

PLOS Biology

Reviewer remarks:

Reviewer #1: The new manuscripts and the replies are satisfactory. 

Reviewer #2: The authors have successfully addressed my comments. The manuscript is much clearer in terms of the novelty of the matched work and delay tasks and the analysis made. There are only some minor issues remaining. 

I am still not convinced about the quadratic relationship between D1 blockade and incentive motivation based on the data of two monkeys presented and I would suggest that this statement is further tuned down. Beyond saying that the relationship tends to be U-shaped in the discussion, it needs to be clear that the relationship between D1 blockade and refusal rate is inferred solely based on 2 monkeys and that there is no statistics supporting its existence for this behavior at the population level. In line 180-181 the authors could say "went clearly up for monkey KN but less clearly so for monkey ST". 

In new analysis, the authors show that refusal rates and RT are correlated, possibly both reflecting motivation. If this is the case, and the effects of D1 blockade on motivation are quadratic, shouldn't the effects of SCH23390 on RT also be quadratic?

What do the authors have in mind when suggesting (on line 334-335) that the effects of D2 blockade on workload are through a distinct motivational process? I find the dissociation in RT/refusal rate correlations after different levels of D2 blockade very intriguing and I believe it deserved further discussion.

Reviewer #3 (John Salamone): The authors did a very good job addressing the comments.

---

## [Editor Report · Decision Letter 3]

27 May 2021

Dear Dr Minamimoto,

On behalf of my colleagues and the Academic Editor, Matthew Rushworth, I am pleased to say that we can in principle offer to publish your Research Article "D1- and D2-like Receptors Differentially Mediate the Effects of Dopaminergic Transmission on Cost/Benefit Evaluation and Motivation in Monkeys" in PLOS Biology, provided you address any remaining formatting and reporting issues. These will be detailed in an email that will follow this letter and that you will usually receive within 2-3 business days, during which time no action is required from you. Please note that we will not be able to formally accept your manuscript and schedule it for publication until you have made the required changes.

As one last minor point, while going through your revised manuscript, we noticed a minor typo on line 497 ('empathize' should be 'emphasize'). You can fix this while addressing the formatting and reporting requests, to come. 

PRESS

Thank you again for supporting Open Access publishing. We look forward to publishing your paper in PLOS Biology. 

Sincerely, 

Lucas Smith, Ph.D. 

Associate Editor 

PLOS Biology